# Viral RNA Interactome: The Ultimate Researcher’s Guide to RNA–Protein Interactions

**DOI:** 10.3390/v16111702

**Published:** 2024-10-30

**Authors:** Wesley A. Hanson, Gabriel A. Romero Agosto, Silvi Rouskin

**Affiliations:** Department of Microbiology, Blavatnik Institute, Harvard Medical School, Boston, MA 02115, USA; whanson@g.harvard.edu (W.A.H.); gabrielromeroagosto@g.harvard.edu (G.A.R.A.)

**Keywords:** RNA-binding protein (RBP), RNA-protein interactome, RNA secondary structure, virus-host interactome, RNA chemical probing

## Abstract

RNA molecules in the cell are bound by a multitude of RNA-binding proteins (RBPs) with a variety of regulatory consequences. Often, interactions with these RNA-binding proteins are facilitated by the complex secondary and tertiary structures of RNA molecules. Viral RNAs especially are known to be heavily structured and interact with many RBPs, with roles including genome packaging, immune evasion, enhancing replication and transcription, and increasing translation efficiency. As such, the RNA–protein interactome represents a critical facet of the viral replication cycle. Characterization of these interactions is necessary for the development of novel therapeutics targeted at the disruption of essential replication cycle events. In this review, we aim to summarize the various roles of RNA structures in shaping the RNA–protein interactome, the regulatory roles of these interactions, as well as up-to-date methods developed for the characterization of the interactome and directions for novel, RNA-directed therapeutics.

## 1. Introduction

RNA viruses are a group of intracellular parasitic microbes that represent one of the largest financial and public health concerns as made evidently clear by the SARS-CoV-2 pandemic. Other than SARS-CoV-2, well-known pathogens discussed in this review include Sindbis virus (SINV), chikungunya virus (CHIKV), influenza A virus (IAV), human immunodeficiency virus (HIV), and members of the Flaviviridae family [1,2,3]. Despite their small genome sizes ranging from 2–30 kb and their limited coding capabilities, these viruses can reprogram cellular processes and dampen immune responses which result in devastating diseases. Their success is largely due to their ability to recruit and repurpose vital RNA-binding proteins (RBPs) central to processes such as transcription, translation, RNA processing, innate immune evasion, host and tissue tropism, packaging, and egress. Given how extensive this topic is, we will primarily focus on some of the more current pieces of work, but there have been many other reviews focusing on distinct aspects of the RNA–protein interactome between viruses and their host cell [4,5,6]. Some of the virally co-opted host RBPs discussed in this review involve SND1-mediating transcription initiation of SARS-CoV-2, Staufen1-enhancing ribosomal recruitment of enterovirus, and Mushashi-1-promoting Zika virus (ZIKV) neurotropism.

Another essential component in an RNA virus toolkit is the use of RNA structure elements for the recruitment of various RBPs and the regulation of key processes. Perhaps one of the most well-researched cases of structure-driven RNA–protein interactions is HIV’s trans-activation response element RNA (TAR) which folds into an essential stem-loop that recruits the trans-activator protein Tat, enhancing transcription of the viral genome [7,8,9]. Recent advances in sequencing and structure probing methods have resulted in a surge of research focusing on how exactly RNA structures play a role in virus biology [10]. Other well-known cases of RNA structure-driven processes include the use of internal ribosome entry sites (IRES) to recruit ribosomes as well as the 5′ and 3′ untranslated regions (UTRs) which facilitate the recruitment of RBPs crucial for maintaining the stability and efficient translation of viral transcripts [11,12,13,14]. A more unique instance of important RNA structures is seen in HIV-1, CHIKV, and the Coronaviridae family, all of which employ stable stem-loops and pseudoknots to induce ribosomal frameshifts that enable the synthesis of additional proteins, further expanding their genetic repertoire [11,15,16,17]. Outside of these well-studied cases, there has been surprisingly very limited work conducted on characterizing the interplay between viral RNA secondary structures and their interactions with host factors, as discussed in this review.

Furthermore, this review will discuss methods used to study RNA–protein interactions as well as RNA structure in the context of viruses. Many current and emerging methods take advantage of advancements in high-throughput sequencing, RNA–protein crosslinking, and advanced proteomics. We discuss classic protein-centric approaches like crosslinking and immunoprecipitation (CLIP) as well as RNA-centric approaches like comprehensive identification of RNA-binding proteins by mass spectrometry (ChIRP-MS) which have been extensively used to study specific RNA–protein interactions or the broader virus–host interactome [6,18]. This review also discusses methods like Dimethyl Sulfate mutational profiling followed by Sequencing (DMS-MaPseq) and Selective 2’-Hydroxyl Acylation analyzed by Primer Extension (SHAPE) used to study RNA secondary structures in viruses [11,19]. Finally, we delve into the exciting new avenues of therapeutics that focus on disrupting RNA structures and the recruitment of RBPs. By synthesizing this body of work, we aim to provide a comprehensive overview of the current understanding of viral RNA–protein interactions and highlight emerging trends and methodologies in the field.

## 2. Viral Co-Option of Host RNA-Binding Proteins to Aid in Genome Replication

There are multiple host proteins that play an essential role in virus replication. The DEAD-box helicases represent a broad family of proteins with diverse functions involved in the replication of many viruses. DDX1 has been demonstrated to bind to the 5’ and 3’ UTRs of the hepatitis C virus (HCV), aiding in the assembly of the replication initiation machinery [20]. Other members of the Flaviviridae family such as Dengue virus (DENV), Japanese encephalitis virus (JEV), and West Nile virus (WNV) have been shown to hijack DDX3 to aid in replication [21,22,23]. However, other work poses DDX3 as an antiviral host factor for DENV, challenging previous work [24]. More recent work has shown DDX5 to re-localize from the nucleus to the cytoplasm and bind Sindbis virus (SINV) RNA, working synergistically with DDX17 to act as a positive regulator of replication [25].

Broad host–virus interactome screens followed by target-specific validation experiments have proven to be a popular and useful way to identify several host factors that aid in replication. SND1, for example, plays a pivotal role early in the SARS-CoV-2 life cycle (~6 hpi) by mediating nascent RNA synthesis [26] (Figure 1B). SND1 is shown to preferentially bind negative-sense RNA and modulate the localization and covalent binding of NSP9 to the viral RNA, aiding in the initiation of negative-sense RNA synthesis. NSP9 is a viral protein with a covalently attached nucleoside monophosphate (NMPylation) thought to be involved with the priming of viral RNA for transcription [27,28,29]. Knockout of SND1 revealed a dysregulation in NSP9 binding and led to 5–10-fold decreased viral RNA levels, highlighting its importance as a host factor for RNA replication. Another group used a similar approach on SINV to identify various host RBPs that colocalize to the viral transcriptional machinery [30]. Among these, Xrn1 was identified as a pro-viral host factor, a surprising find given its well-known function as an antiviral 5′ to 3′ exonuclease [31]. However, Xrn1 is also known to play an essential role in flaviviruses, as discussed ahead.

## 3. RNA–Protein Interactions in Viral RNA Processing

Xrn1 is responsible for generating subgenomic flavivirus RNAs (sfRNAs) as a result of stalling induced by stable pseudoknots structures found in the 3′ UTR of the flavivirus genome [32,36] (Figure 1A). During Zika virus (ZIKV) infection, sfRNAs interact with several host proteins, including DDX6 and EDC3 (RNA decay), PHAX (splicing), and APOBEC3C (nucleic acid editing) [33]. The functional relevance of these interactions remains unclear, but in the case of WNV, the interactions between Xrn1, DDX3, and other host proteins involved in P-body formation are essential for WNV replication [23]. It has not yet been shown, but P-body formation could lead to the generation and accumulation of sfRNAs which correlates with the stabilization of certain short-lived host mRNAs such as FOS and JUN [33]. The exact mechanism through which sfRNAs stabilize RNA transcripts is not known, but other groups have identified Vigilin and RRBP1 as proviral factors involved in the stabilization of ZIKV and DENV transcripts, as evident by the increase in decay rates of genomic RNA when either factor is knocked-out [37].

Other groups have focused on studying co-opted host RBPs repurposed for splicing viral RNA. The adenovirus proteins E1B55K and E4orf6 mediate the ubiquitination of the host splicing factors RALY and hnRNP-C, increasing their binding affinity to viral RNA and promoting efficient RNA processing and splicing [15]. Disruption of this ubiquitination process significantly hampers adenovirus replication, emphasizing the role of RBP modification in the viral life cycle. Coronaviruses like SARS-CoV-2 which do not undergo splicing take a more aggressive approach and disrupt host splicing mechanisms to suppress immune responses [27]. The viral protein NSP16 binds to U1 and U2 snRNAs, interfering with the splicing machinery at the mRNA binding site. Another study focused on HIV-1 revealed the role of RNA structure in determining splicing outcomes [38]. Using DMS-Mapseq, researchers identified alternative conformations of the HIV A3 splice site, which exist in a ~60:40% ratio and influence the splicing efficiency. By introducing synonymous mutations to favor or disfavor certain conformations, they were able to modulate splicing efficiency, demonstrating the functional relevance of RNA structure in viral splicing. This work along with others highlights the importance of RNA structural heterogeneity in recruiting repurposed host factors to benefit the viral life cycle [38].

## 4. Viral Co-Option of Host RNA-Binding Proteins to Aid in Translation

Translation is perhaps the most well-studied process perturbed by RNA viruses, involving the translational shutoff of host transcripts, preferential recruitment of translational machinery, and unique mechanisms such as frameshifting. There are many strategies that viruses employ to take control of the cell’s translational landscape, but SARS-CoV-2′s use of NSP1 is currently one mechanism that has been extensively studied. NSP1 is known for binding to the ribosomal entry channel, blocking host mRNA translation while allowing for the translation of viral mRNAs [39]. Structural studies have provided insights into how NSP1 achieves this selective translation, with the viral 5′ UTR playing a protective role that prevents NSP1 from inhibiting viral translation [39,40,41]. Moreover, NSP1 has also been shown to localize to the nucleolus and bind to pre-rRNA, inhibiting ribosomal RNA biogenesis, though the exact mechanism remains unclear [42]. Other proteins such as NSP8 and NSP9 also play roles in disrupting host protein trafficking by binding to the 7SL scaffold of the signal recognition particle (SRP) complex, further hampering interferon secretion and contributing to immune evasion [43].

Other viruses rely more heavily on the recruitment of host factors to enhance the translation of their own transcripts. In the case of Enterovirus 71 (EV-A71), the host protein Staufen1 (Stau1) was shown to bind to the 5′ UTR of the viral genome, promoting translation by enhancing the recruitment of ribosomes to the IRES located within the 5′ UTR [44]. Knockout of Stau1 resulted in a significant reduction in viral RNA copy number and protein levels, while overexpression rescued these effects, indicating Stau1′s role in facilitating efficient viral translation. The initiation and regulation of translation in viral infections often involve complex interactions between viral RNA elements and host factors. For instance, the Barley Yellow Dwarf Virus (BYDV) relies on the eukaryotic initiation factor 4F (eIF4F) to promote a reorientation of eIF3 binding on the 5′ and 3′ UTRs of its mRNA, facilitating translation [45] (Figure 2A). However, the study primarily utilized an artificial reporter system without assessing the phenotypic effects on the live virus, limiting the conclusions that can be drawn about the in vivo relevance of these interactions.

Frameshifting is another key aspect of viral translation, particularly in the context of viruses like coronaviruses and HIV-1. Shiftless (SFL), a host protein, has been identified as a novel regulator of programmed -1 ribosomal frameshifting (-1 PRF), which is crucial for the synthesis of viral proteins such as the HIV-1 Gag-Pol polyprotein [49]. SFL binds to ribosomes and reduces frameshifting efficiency, with knockdown of SFL leading to increased frameshifting, while overexpression has the opposite effect [50]. Further research into SFL’s mechanism revealed that it can induce premature termination of translation during -1 PRF, likely mediated by ribosome rescue factors such as eRF1-eRF3 [51]. SFL has also been shown to suppress DENV infection and form a ribonucleoprotein complex with LARP1 and PABCP1 [52]. LARP1 and PABPC1 have been identified as antiviral RBPS that interact with SARS-CoV-2 RNA, possibly making them key players in the SFL-driven mechanism to inhibit -1 PRF [53]. Studies have shown that short reporters used in frameshifting experiments, such as those involving the SARS-CoV-2 frameshifting stimulatory element (FSE), do not fully capture the -1 programmed frameshifting efficiencies observed in full-length genomes [54]. Zinc-finger antiviral protein (ZAP-S) is another host factor that has been shown to inhibit SARS-CoV-2 frameshifting by interacting with the FSE of the viral RNA and the ribosome. ZAP-S reduced frameshifting efficiency from 46% to ~26% in an in vitro system, and further analysis suggested that ZAP-S hampers the refolding of the FSE pseudoknot structure, a critical element for frameshifting [55]. However, these effects were not tested on the live virus, and the minimal FSE used in the study may not fully capture the complex interactions occurring in vivo, indicating a need for further investigation. Furthermore, the complexity of FSE refolding likely involves alternative conformations and long-distance interactions that are not fully represented in the minimal FSE constructs used [56]. Other work has focused on using CRISPR screens to identify new host factors that influence frameshifting efficiencies in SARS-CoV-2 [57]. Surprisingly, ribosome recycling factors like DENR, ABCE1, YEA1D1, and ORAOV1 represent some of the top hits for positive -1 PRF regulators. These proteins were individually validated in different coronavirus FSEs, demonstrating an ability to bind the FSE and specifically modulate -1 PRF efficiency in SARS-CoV-1 and SARS-CoV-2 as opposed to other betacoronaviruses such as OC43 and HKU1.

## 5. Viral RNA Structures and Host Protein Interactions in Tissue Tropism and Host Adaptation

Arthropod-borne viruses, also known as arboviruses, have garnered much attention recently due to their increasing prevalence and spread. The warming climate has enabled many of their vector-hosts such as mosquitoes and ticks to rapidly spread and thrive in more parts of the world [58]. Their ability to switch between hosts presents a fascinating avenue of research, and many groups have made many advances in understanding the barriers to dual-host adaptation. Zhang et al. identified the process of genome replication as a key barrier for dual-host adaptation. The study demonstrated that swapping the UTRs between ZIKV and an insect-specific flavivirus altered their genome replication capabilities in different host cells. The ZIKV UTRs enable replication in both insect and mammalian cells, while the insect-specific UTRs restrict genome replication to insect cells. The group identified factors that specifically bind to the ZIKV UTRs such as WTAP, SYNCRIP, and G3BP1, which are crucial for genome replication in mammalian hosts. It was further shown that stem-loop 1 of the ZIKV 3′ UTR is sufficient to enable the insect-specific flavivirus to recruit these factors, but more work is needed to determine its essentiality for replication in mammalian cells [59]. Work performed on Chikungunya virus (CHIKV) also identified the 3′ UTR as a key element for dual-host adaptation. The CHIKV 3′ UTR is composed of an assortment of three different direct repeats (DR) known to be hotspots for recombination and differ across lineages of CHIKV [60,61]. It was shown that the composition of the 3′ UTR directly influences CHIKV’s ability to grow either in mosquito cells or mammalian cells. The variability of the 3′ UTR is due to the RNA polymerase template switching between the sequence homology of DRs, which in turn influences CHIKV’s ability to host-switch. This recombination occurs in viruses of the same lineage but also between viruses of different lineages and is proposed as a model to maintain CHIKV population diversity when swapping hosts. Building on this, Bardossy et al. identified a conserved Y-shaped RNA structure within the 3′ UTR of CHIKV, which enhances replication specifically in mosquito cells, further contributing to the virus’s dual-host adaptability [62]. However, the host factors involved in this re-assortment of the 3′ UTR remain elusive.

There has been limited research on host factors and RNA interactions essential for viral tissue tropism, but one notable study highlighted a unique mechanism by which ZIKV co-opts host factors to facilitate neurotropism. ZIKV is known to target neuronal tissues, causing microcephaly in human fetuses [63,64]. Mushashi-1 (MSI-1), an RBP that regulates the self-renewal of neural stem cells, was found to promote neurotropism in ZIKV [20]. MSI-1 was determined to bind the 3′ UTR of ZIKV, and mutating this binding site attenuated the virus in a MSI-1-dependent manner. Interestingly, this binding site forms an AGAA-tetraloop, and altering this structure abolishes MSI-1 binding, but reverting it back to an AGNN-tetraloop-type structure rescues binding. This AGNN-tetraloop structure is unique to ZIKV and knockout of MSI-1 was specifically detrimental to ZIKV as opposed to other flaviviruses, demonstrating how specific RNA structures can determine tissue tropism.

## 6. RNA–Protein Interactions in Viral Genome Packaging and Virion Assembly

Virion assembly and egress are critical steps in the viral infection cycle, requiring precise mechanisms to distinguish viral from host RNA, as well as genomic from messenger RNA. Many viruses utilize genomic packaging signals, which are small, often structured RNA sequences that preferentially recruit packaging constituents such as capsid proteins. Viruses such as mouse hepatitis virus, Venezuelan equine encephalitis virus, and human immunodeficiency virus rely on RNA secondary structures for efficient and specific packaging of the genome into mature virions [34,65,66].

The structure and recognition mechanisms of these packaging elements vary significantly, even within viruses of the same family [65]. For instance, the well-studied HIV psi (Ψ) element depends on 3D structural interactions of a specific lability [35]. Recent research revealed a [UUUU]:[GGAG] duplex within the Ψ hairpin that linearizes upon recognition by the nucleocapsid domain (NC) of Gag. Experiments showed that stabilizing this sequence via the addition of GC base pairs at the base of the hairpin or replacing G:U wobble base pairs with G:C pairs significantly reduced the NC affinity and packaging efficiency. Destabilization of the Ψ hairpin via removal of the uridine tetramer also resulted in significantly impaired NC binding affinity, indicating that both the structure and its ability to dissociate upon recognition by NC are crucial for efficient binding. Another group recently revealed the importance of sequence heterogeneity within the HIV-1 5′-start site for differentiating between genome packaging and mRNA translation. RNAs transcribed with a single 5′ guanosine (1G) adopt a dimeric structure that covers up the 5′-cap, enhancing NC recognition and packaging. Conversely, RNAs with a 3G leader sequence adopt a monomeric conformation that exposes the 5′ cap, promoting eIF4E binding and translation [67]. These bodies of work highlight the complex regulatory mechanisms underpinning a well-defined packaging signal.

In contrast, hepatitis C virus (HCV) packaging does not rely on a single, well-defined packaging signal. A recent study suggests that HCV packaging may be directed by cooperative effects of multiple low-affinity packaging signals [68]. Using systematic evolution of ligands by exponential enrichment (SELEX), researchers identified HCV core protein N-terminal domain 1 ligands from a library of RNA aptamers and discovered eight evolutionarily conserved putative packaging signals (pPSs) in the HCV genome. Simultaneous disruption of all eight sites significantly reduced extracellular and intracellular infectious titer without affecting replication and translation.

Some viruses recruit host RBPs and RNA-editing proteins to prepare the genome for egress. Hepatitis B virus (HBV) pre-genome (pgRNA) packaging exemplifies this strategy. HBV pgRNA depends on a 5′ RNA structure called epsilon [69,70,71,72]. Recent studies identified an m6A methylated base within epsilon that is necessary for interaction with HBV core protein and efficient pgRNA packaging [70,73]. The host methyltransferase METTL3/14 was found to be responsible for this methylation, and its knockdown resulted in profound reductions in pgRNA packaging and rcDNA synthesis, without affecting viral replication.

RNA–protein interactions involved in genome packaging and virion production are diverse and understudied in many virus families such as Flaviviridae [74] and Picornaviridae [75]. Understanding these interactions is crucial for developing new therapeutic strategies and comprehending viral replication cycles. As packaging and egress are essential for producing infectious particles and propagating genetic material, the strategies adopted to optimize the efficiency of this process are under intense evolutionary pressure and are important for estimating the evolutionary trajectory of pathogens of interest.

## 7. RNA–Protein Interactions in Evasion of Host Immunity

Viral RNA secondary structures and their interactions with RNA-binding proteins play a crucial role in evading host immune responses. These structures, including hairpins, pseudoknots, and internal loops, serve to mask viral RNA from recognition by host immune effectors such as small effector RNAs [76,77]. The recruitment of RBPs can hide double-stranded RNA (dsRNA) motifs that would otherwise be detected by pattern recognition receptors (PRRs) such as RIG-I-like receptors (RLRs) and Toll-like receptors (TLRs) [78,79]. By mimicking host RNA elements, viral RNAs can effectively escape immune surveillance. A recent study provided evidence suggesting that viruses with more highly structured genomes less-potently activate host innate immune sensors compared to those with mostly unstructured genomes [80]. This reduced activation is likely due to the various effector functions conferred by these secondary structures, which facilitate immune evasion and antagonism.

The host antiviral response characterized by the activation of type I interferons leads to the upregulation of interferon-stimulated genes (ISGs). This response further activates the integrated stress response (ISR) and results in the phosphorylation of host eukaryotic initiation factor 2 α (eIF2 α), preventing formation of the ternary initiation complex. This phosphorylation, often mediated by protein kinase R (PKR) or PKR-like endoplasmic reticulum kinase (PERK), culminates in near-global translational shutdown within infected cells [81,82]. However, many viruses have evolved strategies to bypass this translational blockade. For instance, SINV utilizes a stable hairpin structure within its subgenomic RNA to enable non-canonical translation initiation, effectively circumventing the ISR translational blockade [83]. Certain viruses, such as foot-and-mouth disease virus (FMDV) include highly structured internal ribosome entry sites (IRES) [84] (Figure 2B). These complex structures allow cap-independent recruitment of host ribosomes, removing the requirement for a 5′ cap and recruitment to the ribosome by eIF4F [85,86], though they are still reliant on the ternary complex including eIF2α [87]. Unlike other IRESes, the cricket paralysis virus (CrPV) intergenic IRES is capable of translational initiation in the absence of any eukaryotic initiation factors or initiator tRNA, allowing translation to proceed even during antiviral translational shutdown (Figure 2C). This feature of the IRES is facilitated by the ability of pseudoknot I to occupy the ribosomal A (acceptor) site, acting as a surrogate for incoming tRNA [88,89,90,91].

Viruses can encode RBPs that serve to dampen the host immune response by concealing immunostimulatory dsRNA molecules. Research has shown that filovirus protein VP35 binds and sequesters dsRNA in virally infected cells, resulting in reduced stimulation of type I interferon. The immunoregulatory role of VP35 is conserved across filoviruses [92]. Similarly, filovirus VP35 and IAV NS1 have been shown to prevent PKR activating protein (PACT) interactions with RIG-I, PKR, and DICER. PACT binding inhibits dsRNA-mediated type I interferon stimulation by RIG-I and PKR, as well as miRNA processing by DICER [93,94].

Host mimicry is another mechanism by which viral RNAs evade immune sensing. One example of this is co-option of host RNA editors. A recent study showed m6A methylation of human metapneumovirus (HMPV) to be inhibitory of the recognition of viral RNA by RIG-I as well as downstream production of type I interferon [95]. m6A was shown to reduce affinity of RIG-I for viral 5′-triphosphate, reducing recognition of genomic RNA. Additionally, YTH domain containing m6A reader proteins have been shown to bind and sequester modified cytoplasmic RNAs [78], potentially reducing the accessibility of viral sequences by immune sensors. It is worth noting that a recent study [96] has called into question the veracity of certain antibody-based m6A detection techniques, such as those used in this study. However, other modified nucleotides such as 5-methylcytidine, pseudouridine, and others have also been shown to less-potently stimulate RIG-I as well as RNA-sensing TLRs [97]. Epigenetic modifications of viral RNAs represent an important strategy for the evasion of host immune sensors such as RIG-I.

Understanding the role of RNA secondary structures and their interactions with host immunity is essential for the development of antiviral therapies. Targeting these structures with antisense oligonucleotides (ASOs) can disrupt their regulatory functions and expose viral RNA to immune detection. Additionally, modulating host RBPs or RNA decay pathways may enhance the host’s ability to detect and eliminate viral RNA. By elucidating these complex interactions, it becomes possible to develop targeted therapies aimed at disrupting viral RNA–protein interactions, ultimately improving host immune responses and creating novel antiviral strategies.

## 8. Current and Emerging Methods Used to Study Viral RNA Structure and Protein Interactions

Surveying the RNA–protein interactome requires methods capable of identifying RNA secondary structures and RNA–protein interaction sites (Table 1). The study of RNA structure has been transformed by techniques such as DMS-MaPseq which allows for the high-resolution analysis of RNA secondary structure in living cells. DMS-MaPseq is particularly useful for detecting unpaired and structured regions of RNA by methylating accessible adenines and cytosines, which are then read out by sequencing to reveal the RNA’s structural landscape. This technique has been applied genome-wide to HIV-1, SARS-CoV-2, and TGEV [38,54,98]. In HIV-1, DMS-MaPseq revealed that the viral genome adopts multiple conformations that play crucial roles in regulating splicing outcomes, with specific structures correlating with different splicing efficiencies [38]. Similarly, when applied to SARS-CoV-2 and TGEV, DMS-MaPseq uncovered the structural diversity of the FSE within infected cells, identifying secondary structural ensembles and long-distance interactions (LDIs) that may be crucial for the virus’s replication and interaction with host machinery [54,98]. These findings underscore the power of DMS-MaPseq in exploring the dynamic structural landscape of viral RNA in its native cellular environment. DMS-MaPseq is a precise and versatile method for studying RNA secondary structures both in vivo and in vitro, offering distinct advantages over other techniques. Its specificity for Watson–Crick base pairing allows for accurate probing of adenines and cytosines, providing unambiguous structural information. The small size of DMS makes it less sensitive to the presence of proteins, allowing for more accurate in vivo structural assessments. A key advantage of DMS-MaPseq is its ability to be read through without the use of manganese during reverse transcription, which typically increases background mutations in other methods. This allows for the use of high-fidelity polymerases, resulting in more accurate sequencing data [99]. SHAPE is another widely used method for probing RNA secondary structures. It offers the potential to assess all four nucleotides by measuring local nucleotide flexibility. While SHAPE provides valuable insights into RNA structure, its reactivity is indirectly related to base pairing, as it interacts with the sugar moiety. The SHAPE technique uses a few different chemicals, which have distinct abilities to permeate and react in cells and may report on different RNA structure features [100]. Importantly, when comparing gold-standard structures folded in vitro, both DMS and the SHAPE reagent N-methylisatoic anhydride (NMIA) result in similar accuracy when incorporated as chemical constraints in computational modeling [101]. It is important to note that some SHAPE chemicals have been validated less thoroughly and researchers must be cautious when choosing a SHAPE chemical for in-cell work to determine which would best report on the question of interest [100]. The selection of the appropriate SHAPE reagent can significantly impact the quality and relevance of the structural data obtained, especially in complex cellular environments. SHAPE has been used to study various viral RNAs, including those of HIV, Dengue, Zika, and Influenza A [102,103,104,105]. Recent studies have utilized SHAPE to compare the genome structures of different DENV serotypes, revealing how synonymous mutations disrupting conserved RNA structures can impact viral replication, and have employed SHAPE-MaP to report novel regulatory motifs that influence virus–host interactions [19,106]. For instance, SHAPE has been applied to the West Nile virus (WNV) genome, identifying conserved RNA structures across mammalian and insect cells that influence viral fitness differently depending on the host [103]. Additionally, studies on Influenza A virus using SHAPE and DMS-MaPseq have provided structural predictions for all eight genome segments in virions and highlighted the importance of these structures in viral replication and pathogenicity [105,107]. It is worth noting that RBPs can sometimes protect RNAs from SHAPE reagents, which would result in a loss of information in that given interaction site [108]. Other techniques involve psoralen crosslinking as a means to study RNA structure in viruses. One such example is the crosslinking of matched RNAs and deep sequencing (COMRADES) method, applied to map long-range RNA–RNA interactions within the SARS-CoV-2 genome [109]. COMRADES revealed around eight long-distance interactions (LDIs) spanning more than 1 kb, with the largest one being 29.8 kb. This finding is the first to show direct evidence of genome cyclization in vivo; however, the functional relevance for this cyclization has yet to be determined. Another LDI of note is termed the FSE-arch which involves the FSE and spans about 1.5 kb in length, a finding further corroborated by DMS-MaPseq [54,104]. COMRADES also revealed interactions between viral RNA and the U1, U2, and U4 snRNAs, highlighting its versatility for identifying cis and trans RNA–RNA interactions [109].

While these techniques focus primarily on RNA structure, high-throughput techniques like VIRal Cross-Linking and Solid-phase Purification (VIR-CLASP) and comprehensive identification of RNA-binding proteins by mass spectrometry (ChIRP-MS) have become invaluable tools for mapping viral RNA–host protein interactions, offering a detailed view of how viruses hijack host cellular machinery (Table 1). VIR-CLASP, applied to the study of CHIKV, identified several host factors that modulate viral replication, including FASN, an enzyme involved in fatty acid synthesis, crucial for viral replication and egress [110]. VIR-CLASP works by UV crosslinking RNA–protein interactions and enriching transcripts in a sequence-independent manner via SPRI bead purification, followed by quantitative proteomic analysis of the enriched protein interactors. However, a clear limitation of broad RNA-interactome methods like VIR-CLASP is the lack of information acquired from low-abundance transcripts. ChIRP-MS, which uses oligos targeting RNAs of interest, proves to be a more transcript-specific method for identifying RNA–protein interactions. One comprehensive study explored the SARS-CoV-2 RNA interactome across different cell types and multiple RNA viruses, identifying a wide range of RNA-binding proteins (RBPs) that interact with the viral RNA. Significant findings included the identification of pro-viral factors crucial for viral replication and antiviral factors that serve as part of the host’s defense mechanisms. Among these, the mitochondrial protein MRM2 emerged as a potential target for therapeutic intervention due to its role in enhancing viral replication. The study also showed significant changes in mitochondrial morphology post-infection, linked to the virus’s ability to manipulate host cellular energy production and apoptosis pathways. These findings were further validated by CRISPR screens, which targeted the identified RBPs to assess their impact on viral replication, revealing critical insights into how the virus co-opts host proteins to enhance its replication while evading immune responses [43]. Similarly, another study focused on the SARS-CoV-2 and OC43 RNA interactome by using biotin-labeled probes to identify RNA–protein interactions specific to both the genomic and sub-genomic RNAs [53]. Many of the RBPs were shared between both coronaviruses, but there were 14 proteins that were detected to interact with the genomic and sub-genomic RNAs of both viruses. The study validates many known interactors with coronavirus RNA such as TRIM25, ZC3HAV1, PARP12, HDLBP, and SHFL, but also identifies other previously unknown interactors. In a complementary study, the SARS-CoV-2 RNA–protein interactome was further characterized using biotinylated probes to pull down the viral RNA, revealing the involvement of vesicle trafficking proteins, cytoskeleton regulators, and RNA-editing cofactors. This work highlighted the roles of ATP1A1 and ARP2 as important pro-viral host factors, offering additional targets for therapeutic intervention [111]. However, multiple ChIRP-MS probes must be designed as to not disrupt the binding of RBPs to the RNA of interest, and enriched factors must be individually validated. A less disruptive method is comparative RNA interactome capture (cRIC), which, depending on the enrichment approach, can either be an RNA-centric or protein-centric approach to studying RNA–protein interactions in viruses. cRIC was used to study the RNA–protein interactome of SARS-CoV-2, revealing time-dependent changes in the viral RNA interactome during infection as well as during antiviral treatments [18]. This group took a more RNA-centric approach by using oligo (dt) capture as their enrichment method and using quantitative proteomics to study the SARS-CoV-2 interactome, but protein-specific antibodies can be used instead as a way to enrich specific RBPs of interest.

Another clever approach to studying RNA–protein interactions is the yeast three-hybrid (Y3H) screening system, which identifies specific host proteins that bind to viral RNA. This method involves expressing viral RNA in yeast and screening it against a library of human proteins to identify novel interactors, expanding the understanding of host–pathogen interactions. The Y3H system is valuable for identifying single RBPs interacting with an RNA of interest, removing potentially confounding host factors [112]. This system utilizes yeast expressing an RNA of interest which are mated with a library of yeast each expressing a different human protein. The resulting progeny express both the RNA of interest and one human protein, but only the ones with a stable interaction remain viable. This approach was validated on the DENV and Zika sfRNAs, which were screened against 12,000 human proteins. This resulted in nine previously known interactors, one of which was DDX6 as previously mentioned, and sixty new interactors [33,112]. This screening approach is also very useful for studying mutations on either the RNA or protein side that can strengthen or weaken their interaction. The clear caveat is it removes the RNA–protein interaction from its native virus infected-cell context. However, pairing ChIRP-MS, cRIC, and the Y3H screening system would serve as a powerful combination for identifying novel RNA–protein interactors and individually validating them.

Some of the newest and most promising methods for studying RNA structure and RNA-binding proteins (RBPs) include O-MAP, TREX, and SEARCH-MaP. These techniques represent significant advancements in the field, offering novel approaches to unraveling the complexities of RNA biology. O-MAP (oligonucleotide-directed proximity-interactome mapping) is a method designed to identify RNA-interacting proteins, transcripts, and genomic loci within their native cellular context. This technique involves fixing cells and targeting an RNA of interest with a primary DNA probe. The primary probe is then targeted with a secondary probe fused to horseradish peroxidase (HRP), which catalyzes the biotinylation of nearby molecules, including proteins, other RNAs, and DNA. The biotinylated molecules are then purified and analyzed, providing a comprehensive view of the RNA interactome [113]. It is worth noting that O-MAP is capable of biotinylating the proximal interactome of an RNA, making it possible to identify factors that are not specifically binding to the RNA. As an alternative, TREX (targeted RNase H-mediated extraction of crosslinked RBPs) is a cutting-edge method better suited at identifying proteins binding specific RNAs of interest in living cells. This method begins by crosslinking RNA–protein complexes in cells, followed by phase separation to extract these complexes. Antisense oligonucleotides (ASOs) and RNase H digestion are then used to specifically target the RNA of interest, releasing the associated proteins. These proteins are subsequently purified and identified through mass spectrometry, enabling precise mapping of RNA–protein interactions [114]. SEARCH-MaP (Structure Ensemble Ablation by Reverse Complement Hybridization with Mutational Profiling) is a method that facilitates the discovery and quantification of long-range RNA base pairs, particularly in complex RNA genomes like that of SARS-CoV-2. This approach involves using antisense oligonucleotides to target specific RNA regions and detect perturbations in DMS mutational profiles, which indicate long-distance interactions [98].

The current advancements in deep learning have prompted the development of in silico predictive approaches to predict RNA–protein interactions and RNA structure. Tools such as PrismNet and RBNet represent significant advancements in predicting RNA–protein interactions using deep learning models trained on experimental data. PrismNet, trained on icSHAPE and CLIP data, predicts RNA–protein interactions within viral genomes by focusing on highly conserved RNA structures, facilitating the identification of potential RNA–protein binding sites. These predictions have been validated using antisense oligonucleotides (ASOs) to disrupt the identified interactions [115,116]. Similarly, RBNet, trained on iCLIP, eCLIP, and miCLIP data, has shown promise in identifying prominent RNA-binding sites, though challenges remain in accurately modeling the full interaction profiles in complex biological contexts [56]. Perhaps the most renowned deep learning tool is AlphaFold 3 (AF3) which can predict the structures of biomolecular complexes, including those composed of RNA–protein interactions [117]. AF3 was evaluated on ten RNA targets from the Critical Assessment of Structure Prediction 15 (CASP15) and was shown to outperform other computational tools used to predict 3D RNA structure [118,119]. Other approaches like EternaFold, MXFold2, UFold, and eFold represent a diverse set of computational tools used to predict RNA secondary structures, each employing a different approach to inform their model predictions [120,121,122,123]. These language model-based approaches represent a growing trend in computational biology, offering powerful tools to predict and analyze RNA structure and RNA–protein interactions on a large scale, complementing experimental methods and guiding further investigations. These advancements will not only aid in enhancing our understanding of viral replication and pathogenesis but also pave the way for the development of new antiviral strategies.

**Table 1 viruses-16-01702-t001:** Selected methods used to survey RNA structure and RNA–protein interactions.

Technique	Primary Use	Description	Viruses Studied	References
DMS-MaPseq.	RNA structure.	Methylates accessible adenines and cytosines, followed by sequencing to map RNA structure.	HIV-1, SARS-CoV-2, TGEV.	[38,54,98]
SEARCH-MaP.	RNA structure.	Antisense oligonucleotides used to perturb DMS mutational profiles and detect long-distance interactions.	TGEV, SARS-CoV-2, SARS-CoV-1.	[98]
SHAPE-MaP/SHAPE-Seq.	RNA structure.	Probes RNA flexibility using SHAPE reagent, followed by reverse transcription and sequencing.	Influenza A Virus (IAV), HIV-1, DENV, ZIKV, WNV, SARS-CoV-2.	[102,103,104,105,106,107,124]
icSHAPE.	RNA structure.	Chemical probing followed by high-throughput sequencing to map RNA structure in vivo.	ZIKV, IAV, SARS-CoV-2.	[115,125,126,127]
PARIS.	RNA structure.	Maps RNA duplexes in vivo by crosslinking RNA secondary structures and deep sequencing.	Enterovirus D68, ZIKV.	[126,128,129]
COMRADES.	RNA structure and interactions.	Crosslinks and deep-sequences RNA to identify long-range RNA–RNA interactions.	SARS-CoV-2.	[109]
CLIP-Seq.	RNA–protein interactions.	Crosslinking and immunoprecipitation of RNA-binding proteins followed by sequencing.	HIV-1, Enterovirus A71, Brome Mosaic Virus, Venezuelan Equine Encephalitis Virus (VEEV).	[130,131,132,133]
eCLIP.	RNA–protein interactions.	Enhanced version of CLIP that includes size-matched input controls for higher accuracy.	HIV-1, SARS-CoV-2, HCMV.	[43,111,134]
PAR-CLIP.	RNA–protein interactions.	Crosslinks RNA–protein interactions using photoactivatable ribonucleoside analogs followed by sequencing.	HIV-1, IAV.	[135,136,137,138]
RIP-seq.	RNA–protein interactions.	Combines RNA immunoprecipitation with high-throughput sequencing to identify RNA molecules associated with specific proteins.	SARS-CoV-2, HIV-1.	[139,140,141]
CLAMP.	RNA–protein interactions.	Crosslinks mRNA–protein complexes using formaldehyde, captures them with streptavidin beads, and identifies bound proteins by mass spectrometry.	Sindbis Virus, CHIKV, VEEV.	[142,143,144]
VIR-CLASP.	RNA–protein interactions.	Uses 4SU/photo-crosslinking, followed by solid-phase separation and mass spectrometry.	Chikungunya Virus.	[110]
Comparative RNA Interactome Capture (cRIC).	RNA–protein interactions.	Maps RNA–protein interactions across the transcriptome at specific stages of infection.	SARS-CoV-2.	[18]
O-MAP.	RNA–protein interactions.	Proximity-interactome mapping using oligonucleotide-directed probes and biotinylation.	N/A.	[113]
TREX.	RNA–protein interactions.	RNase H-mediated extraction of crosslinked RBPs targeting specific RNA regions in living cells.	N/A.	[114]
ChIRP-MS.	RNA–protein interactions.	Hybridizes RNA of interest with biotinylated probes, followed by mass spectrometry to identify bound proteins.	SARS-CoV-2, HCMV.	[43,145,146]
HyPR-MS.	RNA–protein interactions.	Uses antisense probes to hybridize and isolate specific RNA–protein complexes, followed by mass spectrometry.	HIV-1.	[147]
TUX-MS.	RNA–protein interactions.	Incorporates thiouracil into nascent RNA, crosslinks with proteins, and identifies interactions through mass spectrometry.	Poliovirus.	[148]
Yeast Three-Hybrid (Y3H).	RNA–protein interactions.	Expresses viral RNA in yeast and screens it against a library of human proteins to identify novel interactors.	Dengue virus, Zika virus.	[112]

O-MAP and TREX have not yet been applied to viral targets in the literature as indicated by N/A.

## 9. RNA–Protein Interactions Are Promising Targets for Therapeutics

The RNA–protein interactome of virally infected cells effects a variety of essential functions detailed in the above sections. As such, these interactions represent attractive targets for novel, highly specific therapeutics.

Host effector RNAs, such as miRNAs (microRNAs), play significant roles in regulating the cellular response to viral infections. RNA-based therapeutics can be artificially designed to mimic these regulatory roles to target specific pathogens of interest. miRNAs are a class of effector RNAs which guide the host RNA-induced silencing complex (RISC) to target RNAs which are then either degraded by host endoribonucleases or translationally suppressed. Processing of miRNAs relies on recognition of a ~1000 nt sequence including a 33–39 nt hairpin structure termed a primary-miRNA (pri-miRNA) by host nuclease DROSHA, followed by further processing by Dicer resulting in a mature 20–23 nt miRNA which is capable of guiding the RISC complex [149,150]. The specificity and well-defined biogenesis of miRNAs has led to interest in the engineering of novel miRNAs for the treatment of disease. For instance, a recent study demonstrated the efficacy of an artificial miRNA in specifically reducing Huntingtin mRNA in patient-derived cells [151]. This miRNA was designed using pri-miR-451 as a backbone, allowing the formation of a secondary structure conducive to processing by the RISC complex.

Antisense oligonucleotides (ASOs) are short, synthetic strands of nucleic acids designed to bind to specific RNA sequences through complementary base pairing. The primary mode of action for many ASO antivirals is to induce degradation of viral RNA by forming an RNA/DNA duplex and recruiting RNase H [152,153]. In addition to targeted degradation of viral RNA, ASOs can also disrupt regulatory RNA secondary structure [154] or occlude effector sites, thereby reducing the viral load within the host cells. For instance, certain viruses such as HCV rely on host miRNAs for efficient replication. An miR-122 binding site in the HCV 5′ UTR has been shown to be essential for efficient translation, as well as stabilization of secondary structures essential for replication [155,156]. Phase II clinical trials of several ASOs targeted at miR-122 sequestration showed dramatic reductions in viral titer and clearance of infection in the majority of patients [157,158,159]. These studies also highlight a drawback for ASO therapeutics, however, as further investigation revealed a resurgent population of viruses with a mutated miR-122 binding site, rendering the virus resistant to miR-122 sequestration and effectively expanding virus tropism beyond the liver [160].

A newly emerging class of therapeutics takes inspiration from protein-targeting chimera (PROTAC) therapeutics in the form of RNA-targeting chimeras (RIBOTAC). These systems utilize RNA-targeting small molecules fused to ribonucleases, allowing for the specific degradation of effector RNAs of either viral or host origin. Recent studies have demonstrated effective applications of RIBOTAC constructs in the targeting of the SARS-CoV-2 frameshifting element, as well as disease-associated host miR-21 [161,162,163,164]. Anti-miR-21 therapeutics also highlight the feasibility of drug reprogramming, whereby currently approved therapeutics are surveyed for RNA binding potential and repurposed as specific small molecules amenable to RIBOTAC [165,166].

RNA-based and RNA-targeting therapeutics represent a promising frontier in antiviral therapy. By focusing on the structural perturbation, targeted degradation, and sequestration of effector RNAs, these approaches offer innovative strategies to combat disease including viral infections. Understanding the role of RNA secondary structure in the host–pathogen RNA interactome is crucial for the development of these advanced therapeutic interventions.

## 10. Conclusions and Perspectives

RNA–protein interactions in infected cells play an integral role in the regulation of viral replication and the severity of pathogenesis. RNA–protein condensates as well as chemical modifications of host proteins are important for the efficient replication of several viruses [167,168]. In addition, interactions between viral RNAs and host proteins have been suggested to enhance replication efficiency and even expand viral tropism [22,103,169]. The co-option of host factors such as proteins and ribosomal components is a core strategy for the enhancement of viral translation. Structured RNA elements such as IRESes and host factor recruitment signals have been shown to be essential for the viral replication cycle [44,45]. Beyond replication and translation, RNA–protein interactions contribute to important viral processes such as genome packaging and immune evasion. Targeting these important regulatory RNA structures and RBP-binding sites using ASOs and small molecules represents an exciting new frontier in antiviral therapeutics, with recent studies demonstrating efficient replication inhibition in a variety of viruses [170,171,172,173]. Newly developed methods for surveying the RNA–protein interactome in virally infected cells will be instrumental in further study of these important regulatory roles. In addition to the RNA–protein interactome, structural probing of viral RNAs is needed, as relatively few viruses have been assayed in depth to date [19,174]. In conclusion, the RNA–protein interactome represents an important frontier in viral research. Further research into these dynamic structures and interactions is necessary to more completely understand the viral replication cycle and to allow for the informed design of novel therapeutic approaches.

## Figures and Tables

**Figure 1 viruses-16-01702-f001:**
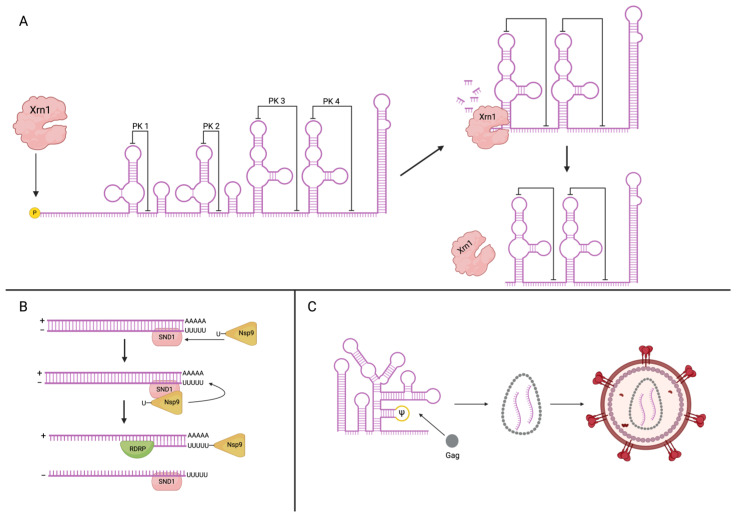
RNA–protein interactions play an integral role in viral replication cycles. (**A**) Xrn1 mediates production of subgenomic flaviviral RNAs (sfRNAs). Host exonuclease Xrn1 recognizes the 5′ terminal monophosphate of uncapped RNAs and degrades the molecule with 5′ to 3′ processivity. Flaviviral (DENV1 shown) 3′ UTRs include a highly structured region featuring several Xrn1-resistant RNA pseudoknots (xrRNAs). Xrn1 degradation is halted at these pseudoknots, resulting in several species of sfRNA with different proviral functions [32,33]. (**B**) SND1 promotes Nsp9 primed antigenome synthesis. Host protein SND1 has been shown to preferentially interact with both the negative-sense SARS-CoV-2 antigenome and the viral protein Nsp9. SND1 remodels Nsp9 occupancy on the viral genome, recruiting it to the 5′ end of the negative-sense antigenome, in proximity with the genomic 3′ UTR, resulting in more efficient priming of antigenome synthesis [26]. (**C**) HIV-1 Psi stimulates genome encapsidation. The HIV-1 Psi element, found in a hairpin loop within the 5′ leader, preferentially recruits the nucleocapsid domain of Gag, allowing for nucleation of the Gag protein to mediate genome packaging [34,35].

**Figure 2 viruses-16-01702-f002:**
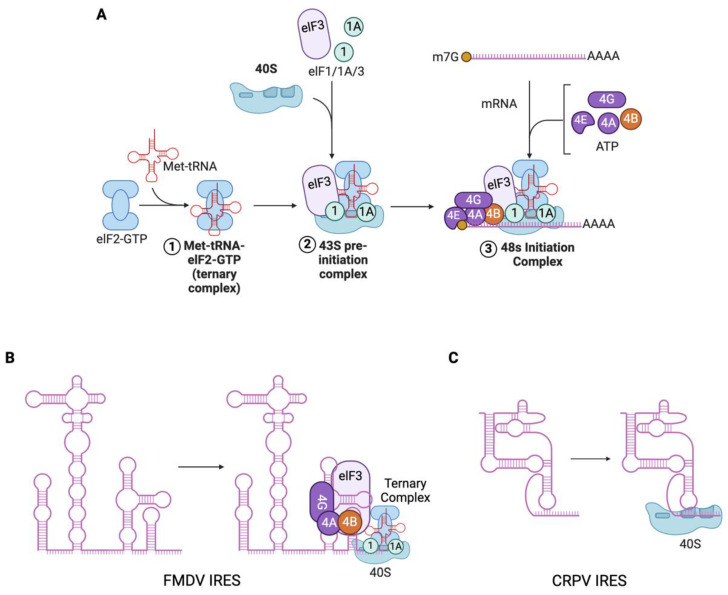
IRES-mediated formation of the translation initiation complex. (**A**) Cellular mRNA translation requires formation of the 43s preinitiation complex. eIF2-GTP complexes with the initiator tRNA (Met-tRNA^Met^_i_) forming the ternary complex. The 40s ribosomal subunit, bound by eIF1, eIF1a, and eIF3, is bound by the ternary complex forming the 43s preinitiation complex. 7-methylguanosine (m7G) mRNA cap is bound by eIF4E which is further bound by eIF4G along with eIF4A and eIF4B. eIF3 binds eIF4G completing formation of the 43s-mRNA complex [46]. (**B**) FMDV IRES forms a translation initiation complex independent of m7G cap. eIF4G binds the FMDV IRES and interacts with eIF4A, eIF4B, eIF3, and the ternary complex, resulting in the formation of the complete initiation complex [47]. (**C**) CrPV IRES directly recruits the 40s ribosomal subunit. CrPV IRES structure features tertiary folds that allow occupation of A, P, and E sites in the 40s subunit. Following recruitment of the 60s subunit, translation is initiated from a GCU codon in the 40s A site [48]. Created with BioRender.com.

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
