# Peer review of "Viral RNA Interactome: The Ultimate Researcher’s Guide to RNA–Protein Interactions"

_viruses, 2024, doi:10.3390/v16111702_

Round 1
Reviewer 1 Report
Comments and Suggestions for Authors
This review has two major sections: first one on the roles of host RBP in virus biology and the second on current approaches identifying RBP-RNA interactomes. The review is interesting- but needs further clarifications and additions. Major comment: Selection of Host RBPs in virus biology by the authors seem to be based on almost random preferences. Major host RBPs are not described or only too briefly:
- Host RNA modification enzymes; such as m6A methylation, poly A addition or pseudouridylation are involved in several viruses, Apobec3 role
- Host RNA helicases are critically involved in HIV, flaviviruses and tombusviruses
L81-101; SARS-Cov-2 N-protein-based biocondensate formation is not relevant here- because it involves viral protein-RNA interaction- all RNA viruses have viral protein-RNA interaction-but not described. So, it is confusing why this section is only on the SARS N protein. Also, the text lacks well-defined host RBPs in N-condensate and SARS-Cov-2 forms double-membrane vesicles for replication compartment, not condensate- although the NP-condensate plays a role. Please modify or omit this confusing part. Within current methods: Please briefly describe the advantages of different approaches/methods in RNA analysis to aim researchers to choose the most suitable approach in their works. Additional comments: Fig. 1B: please replace “antigenome” with “negative-sense” as in the main text (line 109). L79: Replace “transcription” with “replication” L112: Explain “NMPylated” L176: Work on Shiftless was published in “ S. Napthine, C. H. Hill, H. C. M. Nugent and I. Brierley; Viruses 2021 Vol. 13 Issue 7 L186 Reference “90” seems to be cited incorrectly here. L327 What virus is “CrPV”? Table: DS….Correct “viruses studied” column.
Table has no title.
Author Response
Reviewer 1: Thank you so much for your comments, we found your critiques helpful and addressing them has helped enhance this manuscript.
- We have deleted the section regarding SARS-CoV-2 N-condensates and substituted it for a section focused on recent work elucidating the importance of DEAD-box helicases for viral genome replication in flaviviruses and Sindbis virus. The importance of DEAD-box helicases was also briefly talked about in the “RNA-processing” section of the review.
- In the “Evasion of host Immunity” section we have added in a paragraph on host RNA modifications used by viruses to avoid immune sensing.
- In the Current methods’ section, we have added in clarifications regarding the drawbacks and strengths regarding the various methods we discuss, making it clear when a certain method is more suitable.
- We have replaced “transcription” with “genome replication” and replaced “antigenome” with “negative-sense”. NMPylated was further described as the covalent attachment of a nucleoside monophosphate. We further clarified CrPV as Cricket Paralysis virus. Reference 90 was indeed incorrectly cited, and we have now cited the correct reference which is now reference 109; thank you for that catch.
- We read “S. Napthine, C. H. Hill, H. C. M. Nugent and I. Brierley; Viruses 2021 Vol. 13 Issue 7” and have cited it in the main text when discussing Shiftless and its effects on frameshifting modulation.
- We have corrected the table to single space and added some additional viruses studied to the SEARCH-MaP section, but we are a bit unsure as to what exactly was asked to be corrected in the “Viruses Studied” column. We have also added a title to the table “Selected methods used to survey RNA structure and RNA-protein interactions”.

Reviewer 2 Report
Comments and Suggestions for Authors
This manuscript by Rouskin et al. is a very fine review of RNA-protein interactions in viruses. The manuscript focuses particularly on the role of RNA structure, and the methodology for modelling RNA structure and RNA-protein interactions at the viral genome scale. To my knowledge, there is no such review in the literature that would be useful to the community.
I have only a few comments/remarks to make:
l 415: The three yeast hybrids are clearly a very interesting and clever method for identifying RNA-protein interactions, but it was developed in the 1990s and I'm not sure it can still be called innovative.
l 493: Perhaps I'm misunderstanding, but I find this sentence misleading. As far as I know, pri-miRNA can be 1000 nt long or even more, but the recognised hairpin is much smaller.
Translated with DeepL.com (free version)
Author Response
Thank you for indicating these points of confusion. We have rephrased our statement on the YH3 screens and clarified the description of the pri-miRNA.

Reviewer 3 Report
Comments and Suggestions for Authors
This is a well written review on the role of viral RNA structure on regulating viral replication and interactions with host cell machinery. Several examples are given of cellular RNA binding proteins that are usurped by the virus to enhance their expression or dampen immune detection. The examples given are helpful, and the figures are also well done, although I have a few suggestions for improvement. Overall, I think this is a timely review and provide a good summary of methods used to study viral RNA structure and interactions with RNA binding proteins.
Comments for improving the manuscript:
Figure 1C. The figure should be changed to show the full length Gag protein interacting with the psi packaging signal, not the mature NC protein. it is the NC domain, when its part of Gag, that binds to psi to promote viral genomic RNA packaging. The figure legend should be revised to say "allowing for nucleation of the Gag protein to mediate genome packaging"
Page 3, Line 84 should say liquid-like droplets, not liquid droplets, since the droplets are composed of protein and RNA but adopt liquid-like properties when they form biomolecular condensates.
Page 6, second paragraph. A more recent example of the importance of RNA structure could be added to this section by discussing findings from Mike Summers' group showing the importance of heterogeneity of start site of the HIV-1 leader sequence (1G vs 3G) and 5' cap sequestration. This mechanism elegantly explains how unspliced HIV-1 RNA can be differentially used for translation or packaging.
A bit more discussion about the transient nature of viral RNA structure would be helpful to explain the role of dynamic viral RNA statures in complex viral replication cycles. For example, viral RNA can adopt different structures in different cell compartments (nucleus, cytoplasm, ribosome, etc). Also, there is a brief mention of the importance of nascent RNA structure to poliovirus, but this could be expanded since RNA fate can be determined co-transcriptionally.
Author Response
Reviewer 3: We are glad to hear you have enjoyed the review and found your suggestions helpful.
- We have corrected the issue with Fig 1c and now show Gag interacting with the psi packaging signal and have made the proper clarification in the figure legend.
- We have found this suggestion helpful, but we have decided to cut out the phase separation section put entirely as it seemed con using given the lack of recruited host factors. We read and added the suggested Mike Summers work: “Ding, P.; Kharytonchyk, S.; Kuo, N.; Cannistraci, E.; Flores, H.; Chaudhary, R.; Sarkar, M.; Dong, X.; Telesnitsky, A.; Summers, M.F. 5′-Cap Sequestration Is an Essential Determinant of HIV-1 Genome Packaging. Proceedings of the National Academy of Sciences 2021, 118, e2112475118, doi:10.1073/pnas.2112475118.” We found this work fascinating and insightful and will add more depth to our review. Thank you for this suggestion.
- We agree that the transient and dynamic nature of RNA structure is highly important to mediate RNA-protein interactions. However, modern examples in the literature review are scarce and we thought it might complicate an already dense review.
